# Hiding in Plain Sight: Visible Gene Correlations Undermine Single-Cell Representations

**Alon Hacohen, Joseph Bingham, Binyamin Perets & Dvir Aran** *

Technion – Israel Institute of Technology, Haifa, Israel
`{alonhacohen, jbingham, sbp67250}@campus.technion.ac.il`
`{dviraran}@technion.ac.il`

## Abstract

Many single-cell foundation models (scFMs) learn representations of cellular identity through masked modeling of gene expression, yet standard random masking treats genes as independent tokens, a poor match for the modular, co-regulated structure of gene regulatory networks. In this work, we show that this mismatch enables shortcut learning: a model may reconstruct masked genes from locally correlated partners rather than capturing global cellular state, yielding representations that underserve underrepresented cell populations. We introduce CorrMask, a data-driven masking strategy that constructs a gene dependency graph from expression covariance and masks correlated gene groups jointly, forcing the model to rely on higher-order biological context. Evaluating on tissue-specific corpora, CorrMask produces representations that improve cell type annotation, particularly for underrepresented populations, and gene-level generalization, while matching standard baselines with up to $3\times$ less pre-training data. Our results suggest that meaningful single-cell representations require pre-training objectives that respect the dependency structure of the transcriptome.

## 1 Introduction

Single-cell foundation models (scFMs) commonly learn transferable representations from single-cell RNA sequencing (scRNA-seq) data by treating cells as sentences and genes as tokens (Yang et al., 2022; Theodoris et al., 2023; Cui et al., 2024). A central question is what makes these representations *meaningful*, that is, when do they capture genuine cellular identity rather than superficial statistical patterns? We argue that the answer lies not only in architecture or data scale, but in the alignment between the pre-training objective and the biological structure of the transcriptome.

Current scFMs rely heavily on data scaling, yet high-quality biological data is inherently scarce (Chen et al., 2025; Bingham et al., 2025), particularly for rare diseases, transient developmental states, and privacy-preserving settings. Specialized models targeting specific organs or conditions often operate with $\approx 10^6$ cells, orders of magnitude below the scale of generic whole-human models (Theus et al., 2024). This makes it critical to extract maximum learning signal from limited corpora.

We show that a key source of inefficiency lies in the standard random masking objective (Devlin et al., 2019). Gene expression is governed by tight Gene Regulatory Networks (GRNs) where genes operate as functional modules. When a gene and its co-regulated partner are both present in a cell's expression profile, masking one while leaving the other visible creates a trivial reconstruction task, leading the model to learn a local lookup rather than encode the global cellular state. This *shortcut learning* degrades representation quality, particularly for underrepresented cell types that lack the data redundancy to compensate.

While NLP addresses similar issues via span masking (Joshi et al., 2020) or PMI masking (Levine et al., 2021), these solutions fail for scRNA-seq: rank-value encoding or different encoding techniques that rely on expression magnitude ordering (Theodoris et al., 2023) disrupt positional adjacency between co-regulated genes, making adjacency-based masking effectively random with respect to biological structure.

---

*Corresponding author.

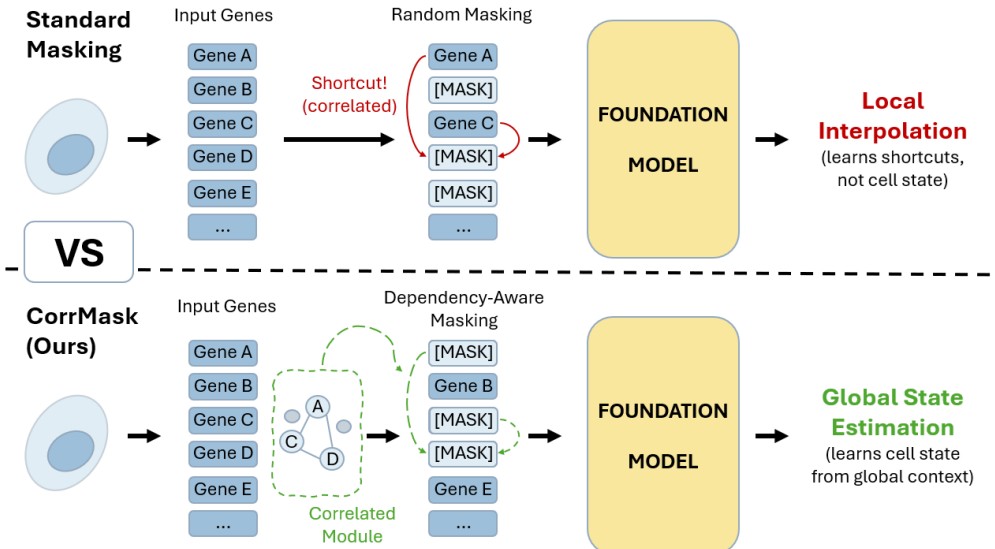

Figure 1: **Overview of CorrMask.** Standard Random Masking treats genes as independent tokens, often masking one gene in a correlated pair while leaving the other visible, allowing the model to cheat via local interpolation. Our approach constructs a latent Dependency Graph based on gene-gene covariance. During pre-training, we mask functional cliques (e.g., a pathway or protein complex), removing local redundancy and pushing the model to infer the missing clique from global cell states.

To address this, we introduce CorrMask, a dependency-aware masking scheme that models the transcriptome as a graph of functional dependencies. During pre-training, we dynamically mask cliques of correlated genes, removing local support and forcing the model to rely on long-range dependencies and global cell-state semantics. Our approach is **data-driven** and **self-contained**: unlike methods relying on external knowledge graphs (Thomas et al., 2019), CorrMask bootstraps dependency structure directly from the training corpus' covariance.

Through scaling experiments across Human Lung, Kidney, Heart, and Pan-Cancer atlases, we show that CorrMask matches standard baselines using up to $\approx 3\times$ less data, with gains concentrated on the long tail of underrepresented cell types, populations where meaningful representations matter most.

**Contributions.** (1) We identify and analyze shortcut learning induced by random masking in single-cell foundation models due to gene regulatory redundancy. (2) We introduce CorrMask, a data-driven dependency-aware masking strategy that removes local gene correlations during pre-training without architectural changes. (3) We show that CorrMask improves sample efficiency by up to $3\times$ and substantially boosts performance on underrepresented cell types across multiple tissues. (4) We demonstrate that correlation-based masking outperforms random and adjacency-based masking under rank-based tokenization.

## 2 RELATED WORK

**Single-Cell Foundation Models.** First-generation models like scBERT (Yang et al., 2022) and Geneformer (Theodoris et al., 2023) introduced masked language modeling for single-cell data, with Geneformer pioneering rank-value encoding. Subsequent models including scGPT (Cui et al., 2024), UCE (Rosen et al., 2023), scFoundation (Hao et al., 2024), and Teddy (Chevalier et al., 2025) explore larger scales and varied architectures, but share reliance on random masking and data scaling.

**Masking Strategies.** SpanBERT (Joshi et al., 2020) masks contiguous spans to force longer-range learning, and PMI-Masking (Levine et al., 2021) targets correlated n-grams. In vision, MAE (He et al., 2022) masks 75% of patches. However, translating these to scRNA-seq is non-trivial: unlike text or images, scRNA-seq is permutation-invariant, and rank-sorting induces adjacency instability.

Tabula (Ding et al., 2025) avoids masking entirely via perturbed inputs, requiring architectural changes. CorrMask defines spans by covariance topology rather than position.

**Structure-Informed Modeling.** Tools like Seurat (Butler et al., 2018) and SingleR (Aran et al., 2019) leverage gene correlations for variable feature selection and cell type annotation, WGCNA (Langfelder & Horvath, 2008) utilizes co-expression patterns for module detection, but scFMs largely ignore this during pre-training. RegFormer (Hu et al., 2025) and scKG-BERT (Li et al., 2025) incorporate external GRNs into attention, but rely on incomplete priors. CorrMask is data-driven, architecture-agnostic, and encodes structure via the masking objective itself.

## 3 SHORTCUT LEARNING IN SCFMS

The effectiveness of masked language modeling depends on whether the masking pattern creates a genuinely challenging reconstruction task. We argue that for single-cell data, standard random masking systematically fails to do so due to the modular, co-regulated structure of gene expression.

**Setup.** Let $\mathbf{s} = (s_1, \ldots, s_L)$ be a rank-encoded sequence of gene tokens for a single cell, obtained by sorting genes by expression magnitude (Theodoris et al., 2023), given the gene vocabulary $\mathcal{G} = \{1, \ldots, G\}$. Under masked language modeling (MLM), a subset $\mathcal{M} \subseteq [L]$ is masked and the model minimizes:

$$\mathcal{L}_{\text{MLM}}(\theta) = \mathbb{E}_{\mathbf{x} \sim \mathcal{D}} \mathbb{E}_{\mathcal{M} \sim p_{\mathcal{M}}} \left[ -\sum_{i \in \mathcal{M}} \log p_\theta(s_i \mid \mathbf{s}_{\bar{\mathcal{M}}}) \right]$$

where $\bar{\mathcal{M}} = [L] \setminus \mathcal{M}$ denotes the visible positions and $p_\theta$ is the model's predictive distribution. Standard practice masks each position independently with probability $p = 0.15$. Ideally, the model should learn to predict masked genes from the global cellular state $C$ (e.g., cell type, differentiation stage), such that $p_\theta(s_i \mid \mathbf{s}_{\bar{\mathcal{M}}}) \approx p(s_i \mid C)$.

**Gene correlations create shortcuts.** Gene expression is governed by regulatory networks where functionally related genes, such as subunits of a protein complex or co-targets of a shared transcription factor, exhibit strongly correlated expression (Langfelder & Horvath, 2008). For two such genes $u$ and $v$ (corresponding to positions $i, j$) with Spearman correlation $|\rho_{uv}| \approx 1$, knowing the expression rank of $v$ nearly determines that of $u$. When $v$ remains visible in $\mathcal{M}$, the model can approximate:

$$p_\theta(s_i \mid \mathbf{s}_{\bar{\mathcal{M}}}) \approx p(s_i \mid s_j)$$

bypassing the global context $C$ entirely. As the model fits this shortcut, the per-token loss reduces to the residual uncertainty of $u$ given $v$, which is negligible for strongly correlated pairs. This is a form of *shortcut learning* (Hermann et al., 2024): the model minimizes training loss through local interpolation rather than by building a representation of cellular identity.

**Shortcuts are pervasive under standard masking.** With a masking rate of $p = 0.15$, any given gene's correlated partner remains visible with probability $1 - p = 0.85$. Span masking (Joshi et al., 2020) offers little relief because rank-value encoding disrupts positional adjacency between correlated genes: two biologically partnered genes may sit far apart in the ranked sequence simply because their expression levels happen to be dissimilar in magnitude. Empirically, we find that correlated gene pairs co-occur within 5-token spans less than 25% of the time across our corpora (Appendix C). Moreover, most genes have multiple correlated partners ($k > 1$), and the probability that *all $k$* partners are simultaneously masked decays as $p^k$, making shortcuts nearly unavoidable even under more targeted strategies like PMI masking (Levine et al., 2021).

**Implications for representation quality.** When shortcuts dominate the training signal, the model learns to attend to local gene neighborhoods rather than encoding cell-level identity. This particularly harms underrepresented cell types, which appear too infrequently for the model to overcome shortcut-driven gradients through sheer repetition. The remedy is to mask correlated gene groups *together*, forcing the model to reconstruct the missing group from the broader cellular context. However, masking *only* correlated groups introduces coverage bias, as large co-expression modules consume the masking budget while singleton genes go unmasked, motivating a hybrid strategy that balances structural masking with random masking.

## 4 CORRMASK

Building on our analysis, we propose CorrMask, a dependency-aware masking strategy operating in two stages: (1) offline construction of a gene dependency graph, and (2) online correlation-guided masking. For the full algorithm, see Appendix A.

### 4.1 DEPENDENCY GRAPH CONSTRUCTION

To construct the dependency graph $\mathcal{G}_{\text{dep}} = (\mathcal{G}, E)$, we sample a subset of $N_{\text{sub}}$ cells from $\mathcal{D}$ and compute a global Spearman correlation matrix based on position-weighted importance scores. For each gene $u$, we select the top-$N_{\text{neigh}}$ candidate neighbors $v$ based on correlation magnitude. We then refine this set using a strict two-step criterion: candidates are retained only if they satisfy a minimum global **Correlation Strength** ($|\rho_{uv}| \geq \rho_{\min}$) and verify **Co-Presence** (co-occurring in $\geq M_{\min}$ cells). For verified neighbors, the final edge is updated based on the Spearman correlation computed strictly on the overlapping cells. This yields a data-driven topology adapted to the specific context.

### 4.2 CORRELATION-GUIDED MASKING

Given an input sequence $x = (g_1, \ldots, g_L)$ and graph $\mathcal{G}_{\text{dep}}$, we compute a total mask budget $B = \lfloor L \cdot p \rfloor$. The mask set $\mathcal{M}$ is constructed via a two-stage process controlled by a **Structural Ratio** $\lambda$ which defines a hard quota for correlation-based masks ($B_{\text{struct}} = \lfloor B \cdot \lambda \rfloor$). We begin with **Structural Expansion**, iterating through shuffled gene seeds. For each seed $s$, we add the seed and its visible neighbors ($g \in \mathcal{N}(s) \cap x$, up to $K_{\max}$) to the mask set $\mathcal{M}$. Then, once $B_{\text{struct}}$ is met or seeds are exhausted, we perform **Random Top-up** to fill the remaining budget $B - |\mathcal{M}|$ by sampling uniformly from unmasked genes ($x \setminus \mathcal{M}$). This ensures the total masking rate $p$ is always met, while $\lambda$ strictly controls the ratio of structural vs. random masking patterns. Finally, we apply the standard 80/10/10 token replacement strategy to the indices stored in $\mathcal{M}$ (Devlin et al., 2019).

## 5 SYNTHETIC VALIDATION

To validate our core hypothesis in a controlled setting, we construct synthetic datasets with modular structure and evaluate whether correlation-based masking recovers it better than random masking.

**Data Generation.** We construct a universe of $n = 1,000$ genes partitioned into $k = 5$ functional modules. Aiming to follow a biologically compelling procedure (Kharchenko et al., 2014; Zappia et al., 2017), each synthetic cell is generated by drawing baseline expression from $\text{Exp}(1)$, activating one module by adding $\mathcal{N}(10, 2)$ signal to all its genes, applying binomial dropout ($p = 0.5$), and retaining only top-$K = 100$ genes (rank-based tokenization).

**Structure Recovery.** We compare two approaches for inferring gene-gene relationships (Figure 2): *Adjacency-based recovery* counts co-occurrences between consecutively-ranked genes (analogous to span masking), and *Correlation-based recovery* counts co-occurrences between all pairs in a cell, regardless of position (based on CorrMask's concept).

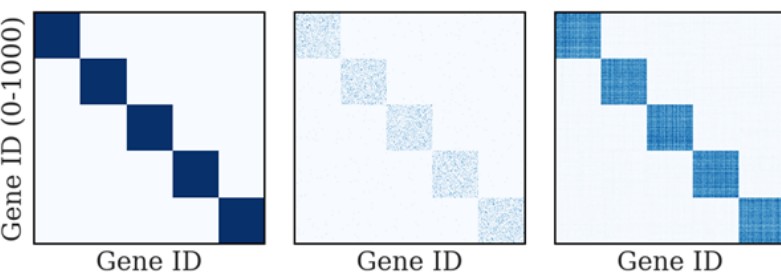

Figure 2: **Modularity Reconstruction.** Ground-truth covariance (left) shows five modules. Adjacency-based recovery (middle) partially recovers structure due to rank instability. Correlation-based recovery (right) reconstructs modular signal strongly.

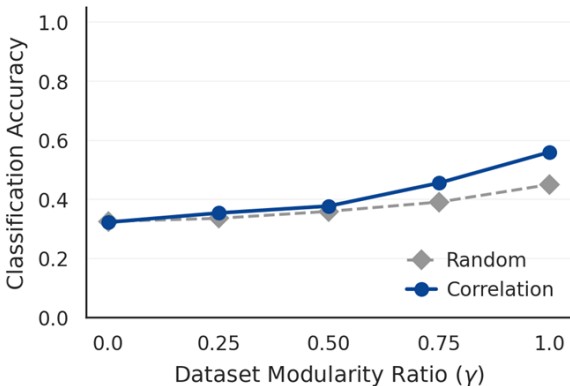

Figure 3: **Modularity-based Performance Gain.** Classification accuracy gap between correlation-based and random masking increases as the data becomes more modular.

Despite both methods operating on identical data, correlation-based recovery reconstructs the structure of the ground-truth covariance matrix better. The adjacency-based method struggles as rank ordering is highly sensitive to small perturbations in expression values, as genes with similar expression levels may swap ranks across samples due to noise fluctuations, destroying any adjacency signal.

**Modularity Sweep.** We then move to construct datasets parameterized by modularity $\gamma \in [0, 1]$ (Figure 3). At $\gamma = 0$, activated genes scatter randomly. At $\gamma = 1$, they confine to ground-truth. We train a lightweight transformer with either random or correlation-based masking, evaluating each on module classification accuracy.

In unstructured regimes ($\gamma < 0.25$), both strategies perform comparably. As modularity increases ($\gamma > 0.5$), correlation-based masking achieves larger gains, reaching $\approx 10\%$ improvement at $\gamma = 1.0$.

## 6 EXPERIMENTS AND RESULTS

### 6.1 EXPERIMENTAL SETUP

**Datasets and Preprocessing.** We utilized four large-scale tissue- or disease-specific scRNA-seq corpora (originally collected by Hui et al. for scGPT's whole-human from the CELLxGENE collection (Cui et al., 2024)) to evaluate performance across diverse biological contexts: **Lung**, **Kidney**, **Heart** and **Pan-Cancer**.

To evaluate sample efficiency, we generated subsets of each corpus at logarithmic scales: **30K, 100K, 300K, and 1M** cells ($\approx$3%, 10%, 30%, 100%). All data preprocessing followed the Geneformer pipeline for reproducibility and to isolate the impact of masking objective from any variances.

**Pre-Training.** We specifically utilize the Geneformer architecture to serve as a baseline to highlight the local redundancy of scRNA-seq and inherent adjacency and rank-instability issues. The only differences between models are the pre-training corpus and the masking strategy used.

**Masking Baselines.** We compare **CorrMask** against three primary baselines: **Random Masking**, the standard agnostic strategy, **PMI Masking**, a structural baseline masking rank-sorted neighbors based on their PMI scores, and **PMI-Rand**, a hybrid of both (50-50% masking each). Unless stated otherwise, CorrMask is hybrid by itself, with $\lambda = 0.5$ found optimal empirically (see Ablations).

**Dependency Graph Construction Details.** As prescribed in Section 4, we construct the dependency graphs per dataset, focusing on the top 2K most highly expressed genes.

**Downstream Tasks.** We fine-tune each pre-trained model on two distinct tasks for benchmarking: *Cell Type Annotation*, classifying at the cell-level using learned embeddings, and *Gene Dosage Sensitivity Prediction*, predicting dosage sensitivity from gene embeddings (Theodoris et al., 2023). These tasks demonstrate the effect of different masking both on the cell and gene levels (i.e. the vector and token levels).

Table 1: **Generalizing on Genes.** AUC for Gene Dosage Sensitivity prediction compared across masking strategies for within/cross tissue.

| TISSUE | TYPE | RANDOM | PMI | PMI-RAND | **CORRMASK** |
|---|---|---|---|---|---|
| LUNG | WITHIN | $0.76 \pm 0.03$ | $\mathbf{0.83 \pm 0.04}$ | $0.77 \pm 0.02$ | $0.82 \pm 0.03$ |
| | CROSS | $0.88 \pm 0.06$ | $0.90 \pm 0.03$ | $0.84 \pm 0.05$ | $\mathbf{0.91 \pm 0.03}$ |
| KIDNEY | WITHIN | $0.76 \pm 0.06$ | $0.80 \pm 0.05$ | $0.75 \pm 0.05$ | $\mathbf{0.82 \pm 0.08}$ |
| | CROSS | $0.88 \pm 0.05$ | $0.85 \pm 0.04$ | $0.83 \pm 0.03$ | $\mathbf{0.89 \pm 0.03}$ |
| HEART | WITHIN | $0.66 \pm 0.08$ | $0.73 \pm 0.05$ | $0.73 \pm 0.11$ | $\mathbf{0.76 \pm 0.04}$ |
| | CROSS | $0.84 \pm 0.03$ | $0.84 \pm 0.02$ | $0.82 \pm 0.06$ | $\mathbf{0.87 \pm 0.04}$ |
| PAN-CANCER | WITHIN | $\mathbf{0.79 \pm 0.11}$ | $0.73 \pm 0.03$ | $\mathbf{0.79 \pm 0.06}$ | $0.78 \pm 0.09$ |
| | CROSS | $0.85 \pm 0.04$ | $\mathbf{0.87 \pm 0.03}$ | $0.85 \pm 0.04$ | $0.86 \pm 0.05$ |

## 6.2 GENE-LEVEL: GENERALIZATION AND ROBUSTNESS

We began by evaluating the model's understanding of fundamental gene regulation through a **Gene Dosage Sensitivity** task. This task, introduced by Geneformer (Theodoris et al., 2023), assesses if the model can predict a gene's sensitivity to dosage changes based on its learned embedding context.

We evaluated generalization on **Within-Tissue**, pre-training a tissue-specific corpus as well as fine-tuning and evaluating on the same tissue, and **Cross-Tissue**, pre-training on a tissue, evaluating on a diverse multi-tissue dataset. We performed 5-fold cross validation, fine-tuning on subsets from all four datasets (see Appendix C).

As shown in Table 1, CorrMask demonstrates superior AUC over baselines across most tissues. This advantage persists in **Cross-Tissue** evaluations, suggesting that by reconstructing functional modules via masking pairs of correlated tokens, CorrMask learns generalizable logic that transfers.

## 6.3 CELL-LEVEL: SCALING AND SAMPLE EFFICIENCY

We next evaluated **Cell Type Annotation**, a primary downstream task for single-cell foundation models (Ovcharenko et al., 2025). We fine-tuned models on the Lung and Kidney cells datasets (with each masking scheme) for 5 epochs on the fine-tuning dataset used by Theodoris et al. (2023), across four orders of magnitude of data scale ($3 \cdot 10^4$–$10^6$). Figure 4 illustrates the scaling behavior.

**Sample Efficiency.** CorrMask consistently shifts the scaling curve to the left. As shown in the figure, CorrMask consistently outperforms random masking across all data scales. On the Lung dataset, a CorrMask model trained on **100K cells** achieves parity with a random model trained on **300K cells**,

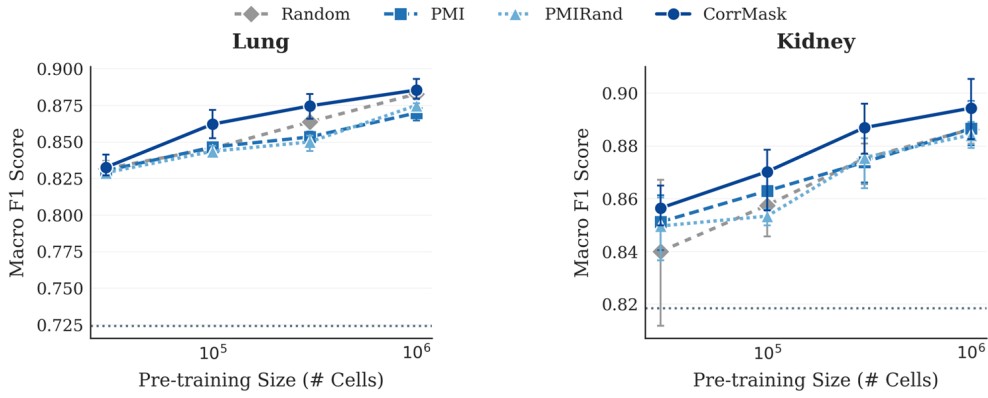

Figure 4: **The Data Multiplication Effect.** Macro-F1 on Cell Type Annotation vs. corpus size (log-scaled) for Lung and Kidney tissues.

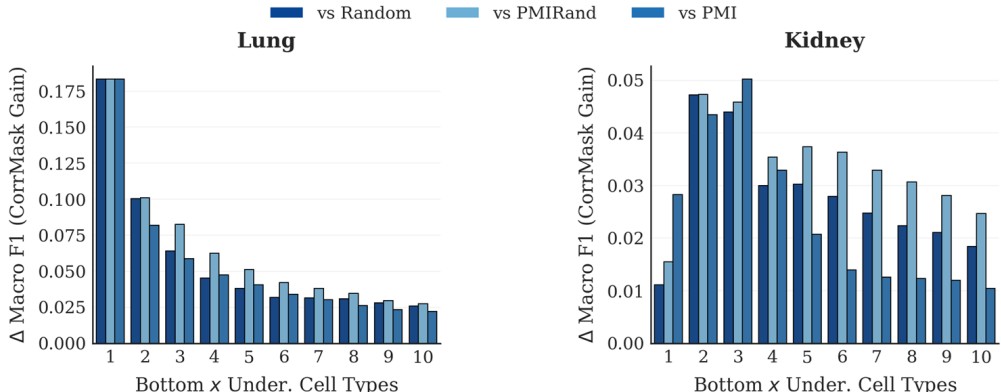

Figure 5: **Rescuing the Long Tail.** Positive change of Macro-F1, $\Delta$Mac-F1 (CorrMask $-$ other method), stratified by cell type frequency.

effectively acting as a $3\times$ data multiplier. Even at the 1M scale, CorrMask outperforms the PMI baseline, confirming that adjacent neighbors addressed with PMI-Masking may not be the correct proxy for dependencies.

## 6.4 RESOLUTION OF UNDERREPRESENTED CELL TYPES

Global metrics often obscure failures on imbalanced datasets that contain cell types that are less abundant and underrepresented cell types. To investigate this, we stratified cell types by their frequency in the training data and calculated the performance gap of the Macro-F1 score for the bottom $X$ number of cell types ($\Delta$F1 = CorrMask $-$ Other Method).

As shown in Figure 5, CorrMask provides a larger boost to underrepresented populations meaning the gains are concentrated in the rarest deciles (e.g., Basal cells in Lung, Fetal Stromal cells in Kidney). This supports our hypothesis that in the absence of massive data redundancy, which abundant cells enjoy, the model relies on CorrMask's structural inductive bias to correctly identify cell identity.

## 6.5 MECHANISTIC AUDIT AND ABLATIONS

**Component Analysis.** To dissect the source of improvement, we compared CorrMask against strict controls on the 100K cells subset of the Lung corpus (Table 2): **Random** Masking, **PMI** Masking, **PMI-Rand** and **CorrMask-100** ($\lambda = 1$), masking neighbors based solely on the correlation graph.

This reveals a hierarchical improvement. Hybrid **CorrMask** achieves a +1.7% gain over random, as well as similar improvements over PMI-based masking, with an even larger gain over pure dependency-aware masking. This suggests that improvement is driven by the *correct* structural awareness, where random masking may provide regularization.

**Sensitivity to Structural Ratio.** To further isolate the impact of the structural inductive bias, we performed a dosage ablation using the same pre-training corpus. We use the parameter $\lambda$ (see Section 4) to control the ratio of structural-to-random masking during pre-training, evaluating five

Table 2: **Component Ablation (100K Lung).**

| METHOD | MACRO F1 | $\Delta$ VS RANDOM |
|---|---|---|
| RANDOM | 0.845 | – |
| CORRMASK-100 | 0.828 | -0.017 |
| PMI | 0.846 | +0.001 |
| PMI-RAND | 0.844 | -0.001 |
| **CORRMASK** | **0.862** | **+0.017** |

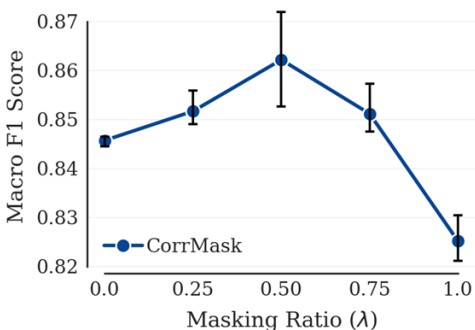

Figure 6: **Structural Ratio Ablation.** Macro-F1 scores for sweep.

regimes ($\lambda \in [0, 0.25, 0.5, 0.75, 1]$). Our results support our hypothesis from Section 3, as $\lambda = 0.5$ was found optimal across regimes (see Figure 6).

**Masking Audit.** We then audited the masking dynamics on the same pre-training corpus to verify our hypothesis (Table 3). We measured the **Conditional Hit Rate**: the probability that a gene's functional partner is masked given that the gene itself is masked. CorrMask achieved **9.32%**, higher than PMI and random ($\approx$**1.94%**). Furthermore, the **Mean Mask Span** for CorrMask was larger as well, confirming our mechanism's effectiveness.

Table 3: **Masking Audit.**

| METHOD | MEAN SPAN | HIT RATE |
|---|---|---|
| RANDOM | 1.08 | 1.93% |
| PMI | 1.08 | 1.95% |
| **CORRMASK** | **1.65** | **9.32%** |

# 7  DISCUSSION AND CONCLUSION

**Representations Reflect Objectives.** Our findings demonstrate that meaningful single-cell representations require pre-training objectives aligned with biological structure. In transcriptomics, genes are co-regulated rather than independent; standard random masking allows models to reconstruct targets via trivial local correlations, yielding representations that encode shortcuts rather than cellular identity. CorrMask corrects this mismatch by masking correlated groups, functioning as an objective-alignment mechanism for structured biological data.

**Sample Efficiency and the Long Tail.** CorrMask improves efficiency by increasing the information density of gradient updates. By removing shortcuts of high correlated genes, each training step better captures higher-order cellular structure. The concentrated gains on underrepresented cell types (Figure 5) are particularly notable: these populations lack the data redundancy that allows abundant cell types to overcome shortcut learning through sheer repetition. This data efficiency enables high-performance modeling in constraint-heavy scenarios, including rare disease characterization and privacy-preserving settings.

**Limitations.** Structural masking relies on estimated dependency graphs, which may be noisy for extremely small corpora. Large gene modules can consume disproportionate masking budget, necessitating the hybrid strategy. Extending CorrMask with dynamic $\lambda$ scheduling, hierarchical dependencies, or joint graph-representation learning are promising directions.

**Broader Relevance.** While evaluated on scRNA-seq, CorrMask applies to any domain with unordered, correlated features, such as proteomics, clinical codes, or even tabular data. More broadly, this work suggests that masking strategy is a first-class design choice for learning meaningful representations, particularly when token independence assumptions break down.

MEANINGFULNESS STATEMENT

This work demonstrates that aligning self-supervised objectives with the modular structure of gene regulatory networks produces more meaningful single-cell representations. By masking correlated gene groups rather than random tokens, our approach pushes models to encode global cellular state instead of exploiting local redundancies. The resulting representations better capture underrepresented cell types, the kind of populations most relevant for disease research and biological discovery. This highlights that representation quality in biology depends not only on data scale but on whether the learning objective respects the underlying biological structure.

LLM USAGE DISCLOSURE

We acknowledge the use of Large Language Models (LLMs) to assist with text editing and grammatical refinement. Additionally, LLMs were used to assist in debugging and optimizing code scripts for data processing, experimentation and visualization. All conceptualization, experimental design, results analysis, and code execution were performed by the authors of this paper, who assume full responsibility for the paper's content.

ACKNOWLEDGMENTS

This work was supported by the Israel Science Foundation (ISF), grant No. 1543/21, and the Miriam and Aaron Gutwirth Memorial Fellowship (2025) awarded to A.H.

We also thank Oren Ploznik and Itai Lavie for their assistance and feedback regarding this work, and the members of the Aran lab for their support and engagement in insightful discussions.

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

## A CORRMASK IMPLEMENTATION DETAILS

### A.1 HYPERPARAMETER SETTINGS

Table 4 summarizes the hyperparameters used in our experiments.

Table 4: CorrMask hyperparameters

| Parameter | Symbol | Value |
|---|---|---|
| *Graph Construction* | | |
| Subset size | $N_{\text{sub}}$ | 100,000 cells |
| Max sequence length | $L_{\text{max}}$ | 2,048 |
| Min co-occurrence | $M_{\text{min}}$ | 200 cells |
| Correlation floor | $\rho_{\text{min}}$ | 0.2 |
| Neighbors per gene | $N_{\text{neigh}}$ | 50 |
| Min gene presence | – | 50 cells |
| *Masking* | | |
| Mask rate | $p$ | 0.15 |
| Max partners per seed | $K_{\text{max}}$ | 5 |

An important property of CorrMask is that the dependency graph can be constructed from any cell population, enabling tissue- or disease-specific adaptation. In our experiments, we construct separate graphs for each pre-training corpus:

**Organ-specific graphs**: For single-organ corpora (e.g., lung, heart, kidney), we build the graph from a 100K cell subset of that organ's data. This captures tissue-specific co-expression programs.

**Pan-tissue graphs**: For diverse corpora spanning multiple tissues (e.g., pan-cancer), the graph captures more general co-expression patterns that hold across biological contexts.

This flexibility allows CorrMask to adapt its inductive bias to the statistical structure of the data without manual curation.

### A.2 COMPARISON WITH PMI-BASED MASKING

An alternative structure-aware approach is PMI (Pointwise Mutual Information) masking, which groups tokens based on sequential adjacency statistics. We implemented PMI masking of bigrams, based on a concept similar to the Naive-PMI introduced by Levine et al. (2021), for comparison:

**PMI Graph Construction**: For each adjacent token pair $(t_i, t_{i+1})$ in the corpus, we compute:

$$\text{NPMI}(u, v) = \frac{\log \frac{P(u,v)}{P(u)P(v)}}{-\log P(u, v)}$$

where $P(u, v)$ is the probability of $u$ and $v$ appearing adjacent, and $P(u), P(v)$ are unigram probabilities.

**PMI Masking**: Sequences are segmented into contiguous spans where adjacent tokens have NPMI $\geq \tau$ (threshold). Whole spans are masked together.

As demonstrated in our validation, PMI-based approaches struggles under the Geneformer constraint (as rank-ordering introduces adjacency noise). Two genes with similar expression levels frequently swap ranks across cells, destroying consistent adjacency signal. CorrMask's co-occurrence approach is robust to this phenomenon.

## A.3 Memory and Runtime Considerations

**Graph storage**: The sparse neighbor list representation requires $O(G \cdot L)$ storage, where $G \approx 25\text{K}$ genes and $L = 50$ neighbors. In practice, this is $< 1\text{MB}$ as a pickled Python dictionary.

**Graph construction time**: The dominant cost is the $O(G^2)$ correlation computation, which we perform block-wise to fit in memory. For $G = 25\text{K}$ and $N_{\text{sub}} = 100\text{K}$, construction takes $\leq 1$ hour on a single CPU.

**Training overhead**: The neighbor lookup can be $O(1)$ per gene using a hash table. The per-sequence masking cost is $O(n \cdot K_{\max})$ where $n \leq 2048$ and $K_{\max} = 5$, which is negligible versus $O(n^2)$ self-attention.

## A.4 Dependency Graph Construction Algorithm

Algorithm 1 provides the complete procedure for constructing the gene dependency graph.

---

**Algorithm 1** Dependency Graph Construction

---

1: **Input:** Corpus $\mathcal{D}$, vocabulary $\mathcal{G}$, $N_{\text{sub}}$, $L_{\max}$, $M_{\min}$, $\rho_{\min}$, $N_{\text{neigh}}$
2: $\mathcal{S} \leftarrow \text{Sample}(\mathcal{D}, N_{\text{sub}})$
3: Initialize $X \in \mathbb{R}^{|\mathcal{S}| \times |\mathcal{G}|}$ with zeros
4: **for** $i = 1$ to $|\mathcal{S}|$ **do**
5:     **for** pos $= 1$ to $\min(|x^{(i)}|, L_{\max})$ **do**
6:         $g \leftarrow x_{\text{pos}}^{(i)}$
7:         $X[i, g] \leftarrow L_{\max} - \text{pos}$
8:     **end for**
9: **end for**
10: **for** each column $j$ in $X$ **do**
11:     $X[:, j] \leftarrow \text{Rank}(X[:, j])$
12: **end for**
13: $X \leftarrow \text{ZScore}(X)$
14: $C \leftarrow \text{Corr}(X)$
15: Initialize $\mathcal{G}_{\text{dep}} = (\mathcal{G}, \emptyset)$
16: **for** each gene $u \in \mathcal{G}$ **do**
17:     candidates $\leftarrow \text{TopK}\Big(\{(v, C[u, v]) : v \in \mathcal{G}, v \neq u\}, N_{\text{neigh}}\Big)$
18:     **for** $(v, \rho_{\text{global}}) \in$ candidates **do**
19:         **if** $|\rho_{\text{global}}| < \rho_{\min}$ **then**
20:             **continue**
21:         **end if**
22:         overlap $\leftarrow \text{CountCoOccurrence}(u, v; \mathcal{S}, L_{\max})$
23:         **if** overlap $< M_{\min}$ **then**
24:             **continue**
25:         **end if**
26:         $\rho_{\text{local}} \leftarrow \text{SpearmanOnOverlap}(u, v; \mathcal{S}, L_{\max})$
27:         Add edge $(u, v, \rho_{\text{local}})$ to $\mathcal{G}_{\text{dep}}$
28:     **end for**
29: **end for**
30: $\mathcal{G}_{\text{dep}} \leftarrow \text{Symmetrize}(\mathcal{G}_{\text{dep}})$
31: **Return** $\mathcal{G}_{\text{dep}}$

---

## A.5 Masking Algorithm

Algorithm 2 describes the online masking procedure applied to each training sequence.

---

**Algorithm 2** CorrMask Masking

---

**Input:** Sequence $x = (g_1, \ldots, g_L)$, dependency graph $\mathcal{G}_{\text{dep}}$, masking rate $p$, partner cap $K_{\text{max}}$, structural ratio $\lambda$

$B \leftarrow \lfloor L \cdot p \rfloor$ {Total Mask Budget}
$B_{\text{struct}} \leftarrow \lfloor B \cdot \lambda \rfloor$ {Structural Quota}
$\mathcal{M} \leftarrow \emptyset$
seeds $\leftarrow$ Shuffle($\{g \in x : \mathcal{N}(g) \neq \emptyset\}$)
**for** $s \in$ seeds **do**
  **if** $|\mathcal{M}| \geq B_{\text{struct}}$ **then**
    **break**
  **end if**
  $\mathcal{M} \leftarrow \mathcal{M} \cup \{\text{pos}(s)\}$
  partners $\leftarrow$ Sample($\mathcal{N}(s) \cap x, K_{\text{max}}$)
  **for** $g \in$ partners **do**
    **if** $|\mathcal{M}| \geq B_{\text{struct}}$ **then**
      **break**
    **end if**
    $\mathcal{M} \leftarrow \mathcal{M} \cup \{\text{pos}(g)\}$
  **end for**
**end for**
**if** $|\mathcal{M}| < B$ **then**
  $\mathcal{M} \leftarrow \mathcal{M} \cup \text{Sample}([n] \setminus \mathcal{M}, B - |\mathcal{M}|)$
**end if**
Apply 80/10/10 replacement to $\mathcal{M}$
**Return** masked sequence, $\mathcal{M}$

---

## B   Synthetic Experiment Details

### B.1   Structure Recovery Experiment (Figure 2)

**Data Generation Process.** We simulate a gene expression matrix with $n_{\text{genes}} = 1000$ genes organized into $k = 5$ equal-sized functional clusters, and $n_{\text{cells}} = 1000$ cells. The generation process for each cell $i$ follows four steps:

**(1) Background expression.** We draw a baseline expression value $x_{ig} \sim \text{Exp}(\lambda = 1)$ for all genes $g$.

**(2) Module activation.** We uniformly sample an active module $m \in \{1, \ldots, k\}$. For all genes $g$ belonging to module $m$, we add a signal $x_{ig} \leftarrow x_{ig} + \epsilon_{ig}, \epsilon_{ig} \sim \mathcal{N}(10, 2)$.

**(3) Technical dropout.** To simulate capture efficiency, we apply a binomial dropout mask $d_{ig} \sim$ Bernoulli($p = 0.5$) such that the observed expression becomes $\tilde{x}_{ig} = x_{ig} \cdot d_{ig}$.

**(4) Rank-based tokenization.** We sort genes by $\tilde{x}_{ig}$ in descending order and retain only the indices of the top-$K = 100$ genes. This yields a sparse sequence $\mathbf{s}_i = (g_1, g_2, \ldots, g_K)$ satisfying $\tilde{x}_{ig_1} \geq \tilde{x}_{ig_2} \geq \cdots \geq \tilde{x}_{ig_K}$.

**Adjacency vs. Correlation-Based Recovery.** Given the corpus of tokenized sequences $\{\mathbf{s}_i\}_{i=1}^{n_{\text{cells}}}$, we construct two distinct gene-gene affinity matrices:

**Adjacency matrix A.** This matrix captures sequential proximity in the list. For each sequence $\mathbf{s}_i = (g_1, \ldots, g_K)$, we increment the entries for all pairs

$$j \in \{1, \ldots, K-1\} : A_{g_j, g_{j+1}} \leftarrow A_{g_j, g_{j+1}} + 1 \quad \text{and} \quad A_{g_{j+1}, g_j} \leftarrow A_{g_{j+1}, g_j} + 1$$

**Co-occurrence matrix C.** This matrix captures set membership regardless of position. For each sequence $\mathbf{s}_i$, we increment the entries for all unique pairs $j < \ell$ appearing in it:

$$C_{g_j, g_\ell} \leftarrow C_{g_j, g_\ell} + 1 \quad \text{and} \quad C_{g_\ell, g_j} \leftarrow C_{g_\ell, g_j} + 1$$

## B.2 Modularity Sweep Experiment (Figure 3)

**Parameterized Data Generation.**    We generate datasets with $n_{\text{samples}} = 1{,}500$, $n_{\text{genes}} = 60$, and $n_{\text{modules}} = 3$. The coherence of gene activation is controlled by a modularity ratio $\lambda \in [0, 1]$. The process then proceeds in four steps:

**(1) Background.** We draw independent baseline expression values for all genes in the genome: $x_{ig} \sim \text{Exp}(0.1)$.

**(2) Module activation.** For each sample, we uniformly sample an active module $m \in \{1, 2, 3\}$. This index serves as the classification label for the sample, $y_i = m$.

**(3) Signal assignment.** We introduce a signal perturbation $\delta \sim \mathcal{N}(10, 0.5)$. With probability $\lambda$ (*coherent activation*), we add $\delta$ to all genes strictly belonging to module $m$. Conversely, with probability $1 - \lambda$ (*incoherent activation*), we add $\delta$ to a random subset of $|m|$ genes selected across the entire genome.

**(4) Tokenization.** Finally, we convert the continuous expression values into discrete ranks within each sample: $\text{token}_{ig} = \text{rank}(x_{ig})$, where $\text{rank} \in \{0, \ldots, n_{\text{genes}} - 1\}$.

This construction ensures that at $\lambda = 0$, the activated genes provide a classification signal but are not organized into recoverable modules, while at $\lambda = 1$, the data exhibits perfect modular structure.

**Model Architecture.** We use a lightweight transformer architecture with a vocabulary size of $n_{\text{genes}} + 2$ (comprising gene tokens, [CLS], and [MASK]), an embedding dimension of 64, 4 attention heads, 2 encoder layers, a feedforward dimension of 128, and learned positional embeddings. Each input sequence is prepended with a [CLS] token. The model is trained for 50 epochs with Adam optimizer (lr=$10^{-3}$), batch size 64, and 25% masking rate.

**Masking Strategies.**    We compare two alternative masking schemes designed to test the model's ability to recover structural information:

**Random masking.** We uniformly sample 25% of the gene positions (excluding the special [CLS] token) to replace with a mask token. This serves as a baseline where no structural bias is introduced into the missingness pattern.

**Correlation-based masking.** This strategy utilizes the ground-truth covariance matrix $\Sigma$, where $\Sigma_{g,g'} = 1$ if genes $g$ and $g'$ belong to the same module. We initialize an empty mask set $\mathcal{M}$ and populate it iteratively until $|\mathcal{M}| \geq 0.25 \times n_{\text{genes}}$. Each iteration proceeds as follows:

First, we sample a seed gene $g$ uniformly and identify its set of correlated partners $\mathcal{P}_g = \{g' : \Sigma_{g,g'} > 0.5, g' \neq g\}$. From this set $\mathcal{P}_g$, we select up to 8 random partners to add to $\mathcal{M}$. With probability 0.5, we also add the seed gene $g$ itself. Finally, if the mask set $\mathcal{M}$ is still below the target size after the iterative process, we fill the remaining slots with randomly selected genes.

Both strategies use BERT-style masking: 80% of masked positions receive [MASK], 10% receive a random token, and 10% remain unchanged.

**Evaluation Protocol.** We split data into 800 training samples for MLM pre-training, and 500 test samples. After pre-training, we freeze the transformer and extract [CLS] embeddings for all samples. A logistic regression classifier (max 2000 iterations) is trained on training embeddings to predict the active module label $y_i \in \{0, 1, 2\}$. For each modularity level $\lambda \in \{0.0, 0.25, 0.5, 0.75, 1.0\}$, we run 5 independent trials with different random seeds, and report mean classification accuracy on the held-out test set.

## B.3 Connection to Real scRNA-seq Data

Beyond mimicking the modularity of gene programs and the rank-based tokenization design, our synthetic experiments abstract key properties of real scRNA-seq, such as dropout effects where truly expressed genes fail to be detected which we simulate this using the binomial dropout), or sample heterogeneity where different cells activate different programs which we simulate using the random module activation per sample (Kharchenko et al., 2014; Zappia et al., 2017). This synthetic validation thus provides a controlled environment to test if correlation-based masking can leverage biological structure under conditions closely matching the real data regime.

## C EXPERIMENTAL DETAILS

### C.1 PRE-TRAINING CONFIGURATION

Table 5: Pre-training hyperparameters.

| PARAMETER | VALUE |
|---|---|
| *Architecture* | |
| Model type | Geneformer (BERT-based) |
| Hidden dimension | 256 |
| Feedforward dimension | 512 |
| Attention heads | 4 |
| Layers | 6 |
| Max sequence length | 2,048 |
| Vocabulary size | $\approx$25,500 |
| *Training* | |
| Optimizer | AdamW |
| Learning rate | $1 \times 10^{-3}$ |
| LR schedule | Linear decay |
| Warmup steps | 10,000 |
| Weight decay | 0.001 |
| Epochs | 3 |
| Batch size | 12 |
| Dropout | 0.02 |
| *Masking* | |
| Mask rate | 15% |
| [MASK] replacement | 80% |
| Random replacement | 10% |
| Unchanged | 10% |
| *CorrMask* | |
| Structural ratio $\lambda$ | [0.0, 0.25, 0.5, 0.75, 1.0] |
| Max partners per seed | 5 |
| *Compute* | |
| GPU | NVIDIA V100 (32GB) |
| Training time (1M) | $\approx$16–18 hours |

Table 5 shows our pre-training hyperparameters. As mentioned in Section 6, We utilized four large-scale tissue- or disease-specific corpora for our pre-training experiments at scales of up to 1M cells per corpus. As this it true for most tissues, the entire kidney corpus does not contain 1M cells, therefore we used it in its entirety ($\approx$814K cells).

### C.2 DOWNSTREAM TASKS PROTOCOLS

We strictly adhere to the fine-tuning and evaluation protocols established by Theodoris et al. (2023) to ensure fair benchmarking. For both downstream tasks, Cell Type Annotation and Gene Dosage Sensitivity Prediction, we fine-tuned the pre-trained foundation models on task-specific datasets as detailed below.

**Cell Type Annotation.** For this task, we utilized the lung and kidney cells from the human cell atlas benchmarking dataset, made publicly available by the authors of scDeepSort (Shao et al., 2021). Following their method, we keep cells with more that 5% of the total cells in each tissue and proceed with a similar procedure for splitting data into training and evaluation (16 cell types across $\approx$33K cells for lung; 15, $\approx$45K for kidney). Models were fine-tuned for 5 epochs using a classification head to predict cell identity labels.

**Gene Dosage Sensitivity Prediction.** Following the methodology of Theodoris et al. (2023), We fine-tuned the models on distinct subsets of the tissue corpora described in Section 6, ensuring no overlap with the pre-training data. We implemented two evaluation settings:

Table 6: **Complete CTA Results.** Macro-F1 scores on Lung and Kidney tissues across scales.

| METHOD | 30K | 100K | 300K | 1M |
|---|---|---|---|---|
| | | LUNG | | |
| RANDOM | $0.831 \pm 0.005$ | $0.845 \pm 0.001$ | $0.864 \pm 0.008$ | $0.883 \pm 0.001$ |
| PMI | $0.830 \pm 0.004$ | $0.846 \pm 0.001$ | $0.853 \pm 0.005$ | $0.870 \pm 0.004$ |
| PMI-RAND | $0.829 \pm 0.003$ | $0.844 \pm 0.004$ | $0.850 \pm .0007$ | $0.875 \pm 0.001$ |
| **CORRMASK** | $\mathbf{0.833 \pm 0.008}$ | $\mathbf{0.862 \pm 0.010}$ | $\mathbf{0.875 \pm 0.009}$ | $\mathbf{0.886 \pm 0.007}$ |
| | | KIDNEY | | |
| RANDOM | $0.840 \pm 0.028$ | $0.858 \pm 0.012$ | $0.875 \pm 0.009$ | $0.886 \pm 0.007$ |
| PMI | $0.851 \pm 0.010$ | $0.863 \pm 0.010$ | $0.874 \pm 0.008$ | $0.887 \pm 0.009$ |
| PMI-RAND | $0.850 \pm 0.012$ | $0.853 \pm 0.004$ | $0.876 \pm 0.010$ | $0.884 \pm 0.005$ |
| **CORRMASK** | $\mathbf{0.856 \pm 0.008}$ | $\mathbf{.0870 \pm 0.013}$ | $\mathbf{0.887 \pm 0.010}$ | $\mathbf{0.894 \pm 0.011}$ |

*Within-Tissue Generalization:* The model was fine-tuned on a specific tissue (e.g., Lung) and evaluated on a held-out validation set of 10,000 cells from the same tissue.

*Cross-Tissue Generalization:* The model was fine-tuned on a single tissue (e.g., Lung) and evaluated on a disjoint set of 50,000 cells sampled from the diverse, multi-tissue reference corpus introduced by Theodoris et al. (2023).

In all experiments, fine-tuning was performed for 1 epoch to adapt the gene embeddings for the dosage sensitivity classifier.

## C.3 COMPLETE CELL TYPE ANNOTATION RESULTS

Table 6 shows our full results across our Cell Type Annotation experiments.

## C.4 CROSS-TISSUE TRANSFER

In an additional transferability evaluation, we use the pre-trained model on Pan-Cancer corpus (1M) and fine-tune it to identify lung and kidney cell types (See Table 7). CorrMask maintains its advantage even when pre-training and evaluation domains differ, indicating more transferable representations.

## C.5 COMPARISON WITH LARGE-SCALE GENEFORMER

For the cell type annotation task, we compared our specialized models (pre-trained on $\leq$ 1M cells) against the original Geneformer (Theodoris et al., 2023), which was pre-trained on a corpus approximately $30\times$ larger ($\approx$ 30M cells) using random masking. The original model achieves Macro-F1 scores of 0.905 (Lung) and 0.909 (Kidney) on average, representing a small gain ($< 0.02$) over our CorrMask results. This suggests that specialized models can achieve performance competitive with massive-scale pre-training despite using significantly less data.

## C.6 CORRELATED GENES WITHIN SPANS

To empirically validate our claim from Section 3 that adjacent genes are not necessarily correlated, we sample 10K cells across all pre-training corpora and calculate the percentage of spans containing a pair of correlation genes (up to spans of 5-grams). In Table 8, we see that for these datasets, a

Table 7: **Cross-Tissue CTA.**

| METHOD | LUNG (MAC-F1) | KIDNEY (MAC-F1) |
|---|---|---|
| RANDOM | $0.719 \pm 0.005$ | $0.802 \pm 0.001$ |
| **CORRMASK** | $\mathbf{0.732 \pm 0.003}$ | $\mathbf{0.808 \pm 0.001}$ |

Table 8: **Correlated Neighbors within 5-grams.**

| LUNG | KIDNEY | HEART | PAN-CANCER |
|------|--------|-------|------------|
| 9.34% | 18.16% | 14.67% | 23.01% |

correlated gene (i.e., possible functional partner using correlation as a proxy) exists inside the 5 closest tokens on either side of the targeted gene at a rate from $< 10\% - < 25\%$ of the time.

## C.7 TRAINING PROTOCOL

To rigorously evaluate sample efficiency, we adopted a compute-constrained protocol. All models were pre-trained for a fixed budget of 3 epochs using the AdamW optimizer using max LR $1e - 4$ and 10k warmup steps, similarly to Geneformer protocol, but utilized a single NVIDIA V100 (32GB) GPU and an effective batch size of 12 to simulate real-world, scientific, low-resource constraints.

