# OpenReview forum: "Hiding in Plain Sight: Visible Gene Correlations Undermine Single-Cell Representations"
_ICLR.cc/2026/Workshop/LMRL — ICLR 2026 Workshop LMRL Poster_

### Official Review · Reviewer_K3He · 2026-02-21

**Rating:** 6
**Confidence:** 4

**Review:**

Pros:
- The paper presents a clear and compelling hypothesis that random masking induces shortcut learning due to gene co-regulation, and this insight is well supported by both theoretical reasoning and empirical evidence.
- The proposed CorrMask method is conceptually simple, architecture-agnostic, and easy to integrate into existing masked modeling pipelines, which increases its practical applicability.
- The inclusion of synthetic experiments to validate the modularity hypothesis strengthens the causal argument and helps isolate the mechanism behind the observed improvements.


Cons:
- The empirical improvements, while consistent, are relatively modest in magnitude, and it is unclear whether they justify the additional complexity of constructing dependency graphs in all practical settings.
- The evaluation is largely limited to a single backbone architecture (Geneformer), making it difficult to assess how broadly the gains generalize across different model families.
- The dependency graph relies on correlation estimates that may be noisy or dataset-specific, and the robustness of the method to graph estimation errors is not thoroughly explored.
- Comparisons to stronger or larger-scale baselines are limited, and the claim of sample efficiency would be more convincing with compute-matched comparisons or scaling law analysis.
- The theoretical discussion of shortcut learning remains mostly qualitative, and a more formal analysis (e.g., information-theoretic framing) would strengthen the argument.
- Some methodological details, such as the choice of hyperparameters for graph construction and masking quotas, appear empirically tuned and lack deeper justification.

---

### Official Review · Reviewer_jZga · 2026-02-24
**Mitigating Shortcut Learning in scFMs with Correlation-Guided Masking**

**Rating:** 8
**Confidence:** 3

**Review:**

CorrMask mitigates shortcut learning in scFMs by jointly masking covariance-derived gene cliques instead of independent tokens. Improves sample efficiency (~3×), gene-level generalization, and rare cell-type performance with minimal overhead.

### Pros
- Masking strategy explicitly aligned with gene co-regulation structure.
- Architecture-agnostic, simple masking modification.
- Strong gains on underrepresented populations.

### Cons
- No ablation on dependency graph noise sensitivity or graph construction hyperparameters.
- Empirical validation restricted primarily to Geneformer backbone.

---

### Official Review · Reviewer_GUCU · 2026-02-24
**Convincing masking strategy needing evaluation beyond Geneformer**

**Rating:** 7
**Confidence:** 4

**Review:**

The work introduces a token masking strategy aiming to minimize shortcut learning artefacts caused by gene correlations during single-cell RNA-seq foundation model pretraining. When a masked gene's correlated partners remain visible, the model can predict the masked target from those partners alone rather than learning to infer it from the broader expression profile. The masking strategy relies on a gene dependency graph, constructed from gene correlations estimated from data. Rather than masking genes uniformly at random, the approach jointly masks genes and their correlated neighbors in the dependency graph, while also incorporating uniform random masking to avoid coverage bias toward genes in large modules.

The masking strategy is benchmarked against alternative masking strategies using Geneformer on four datasets, focusing on cell-type annotation and gene dosage sensitivity downstream tasks for evaluation. The experimental design is thorough, with experiments overall supporting the paper's claims for foundation models using ranking-based encodings. Some experiments are less informative, e.g., the masking audit (Table 3) confirming the algorithm co-masks correlated genes more often than random masking, which is expected by construction. The sensitivity analysis over $\lambda$ is useful, showing that performance is robust across a range of values while supporting the need for a hybrid approach.

While the method is overall clear, it needs some tightening. In particular, it should be clarified that $i, j$  correspond to the positions of genes $u, v$. $\bar{M}$ should also be explicitly introduced in the text. Finally, the 80/10/10 token replacement strategy mentioned in Section 4.2 should include a citation to Devlin et al. (2019).

Overall, the work convincingly adapts pretraining strategies originally developed for natural language to a single-cell setting, and would be a strong contribution to the workshop. It describes a simple protocol acting as a replacement for uniform random masking in foundation model pretraining, showing modest but consistent improvements over baselines.

The clearest limitation of the paper is that all evaluation focuses on Geneformer, leaving unclear whether the approach benefits foundation models using expression-based encodings (e.g., scGPT or scFoundation), more closely reflecting the current state of the art. An explicit justification for selecting Geneformer and a discussion of expected benefits for expression-based models could have strengthened the paper, especially since shortcut learning may be even more severe in that setting, where correlated genes have near-linear relationships making reconstruction trivial.

---

### Meta-Review · Area_Chair_7qNw · 2026-02-25

**Recommendation:** Accept (Poster)
**Confidence:** 4

**Metareview:**

Accept.

---

### Decision · Program_Chairs · 2026-03-02

**Decision:**

Accept (Oral)

**Comment:**

Please see the meta-review.